# Unmasking the Antifungal Activity of *Anacardium occidentale* Leaf Extract against *Candida albicans*

**DOI:** 10.3390/jof10070464

**Published:** 2024-06-29

**Authors:** Luis F. Quejada, Andrea X. Hernandez, Luis C. Chitiva, Claudia P. Bravo-Chaucanés, Yerly Vargas-Casanova, Robson X. Faria, Geison M. Costa, Claudia M. Parra-Giraldo

**Affiliations:** 1Unidad de Proteómica y Micosis Humanas, Grupo de Enfermedades Infecciosas, Departamento de Microbiología, Facultad de Ciencias, Pontificia Universidad Javeriana, Carrera 7 No. 43-82, 110231 Bogotá, Colombia; lquejada@javeriana.edu.co (L.F.Q.); claub06@gmail.com (C.P.B.-C.); y.vargasc@javeriana.edu.co (Y.V.-C.); 2Grupo de Investigación Fitoquímica Universidad Javeriana (GIFUJ), Departamento de Química, Facultad de Ciencias, Pontificia Universidad Javeriana, Carrera 7 No. 43-82, 110231 Bogotá, Colombia; hernandez_a@javeriana.edu.co (A.X.H.); chitival@javeriana.edu.co (L.C.C.); modesticosta.g@javeriana.edu.co (G.M.C.); 3Laboratório de Toxoplasmose e outras Protozooses, Instituto Oswaldo Cruz, Fundação Oswaldo Cruz-FIOCRUZ, Rio de Janeiro 21045-900, RJ, Brazil; salvador@ioc.fiocruz.br; 4Departamento de Microbiología y Parasitología, Facultad de Farmacia, Universidad Complutense de Madrid, Plaza Ramón y Caja S/N, 28040 Madrid, Spain

**Keywords:** *Anacardium occidentale*, *Candida albicans*, *Candida auris*, antifungal resistance, invasive, antifungal treatment

## Abstract

Invasive fungal disease causes high morbidity and mortality among immunocompromised patients. Resistance to conventional antifungal drugs and the toxicity associated with high doses highlight the need for effective antifungal therapies. In this study, the antifungal potential of the ethanolic extract of *Anacardium occidentale* (Cashew Leaf) leaves were evaluated against *Candida albicans* and *C. auris*. The antifungal activity was tested by the broth microdilution method and growth kinetic test. To further explore its antifungal action mode, spectrofluorophotometry, confocal microscopy and scanning and transmission electron microscopy were performed. Additionally, heterozygous knockout strains associated with resistance to oxidative stress were included in the study. We found that *A. occidentale* could inhibit the proliferation and growth of *C. albicans* at concentrations of 62.5 and 125 μg/mL. The doubling time was also drastically affected, going from 2.8 h to 22.5 h, which was also observed in *C. auris*. The extract induced the accumulation of intracellular reactive oxygen species (ROS), resulting in endoplasmic reticulum stress and mitochondrial dysfunction, while it did not show cytotoxicity or hemolytic activity at the concentrations evaluated. Our work preliminarily elucidated the potential mechanisms of *A. occidentale* against *C*. *albicans* on a cellular level, and might provide a promising option for the design of a new treatment for invasive candidiasis.

## 1. Introduction

Annually, invasive fungal diseases (IFDs) are responsible for the admission of more than one million patients to health services worldwide. In addition, they are associated with high mortality rates of ~40–60% [1,2,3]. Among them, invasive candidiasis (IC) is the most frequent, and *C. albicans* is still the predominant etiological agent, being isolated in 43.6% of cases [4,5]. In parallel, *C. auris* has been described as a multidrug-resistant yeast that is difficult to diagnose. It was first described in 2009 in Japan, and has spread worldwide. In 2016, an alert was issued by the Centers for Disease Control and Prevention (CDC) for the mandatory notification of cases attributed to *C. auris* [6,7,8]. It is estimated that, to date, between 3000 and 4000 cases have been reported [9,10]. In South America, the first cases were reported in Venezuela and Colombia between 2012 and 2013, according to prospective studies [6,10]. An increase in cases was detected in several hospitals around the world, with the most recent outbreak being in a hospital in Brazil, forcing the closure of the medical center [11]. Taken together, due to the increase in cases, the limited number of commercial antifungals, the frequent cases of resistance and the toxicity associated with them, it is important to highlight the need to study new effective antifungal therapies^.^ [1,12,13].

Natural products (NPs) have greatly inspired the search for and development of new therapeutic agents. Over the last 40 years, almost half of the drugs approved by the U.S. Food and Drug Administration (FDA) have been based on NPs, either isolated, derived from them, or with modified molecules [14,15]. From this perspective, plants remain a great resource, since it is estimated that a large percentage of the world’s population still uses plants or traditional botanical preparations for the treatment of diseases [15]. However, only 0.8% of the drugs approved by the FDA are considered botanical NPs or botanical drugs, probably because the category was newly introduced in 2012 [14]. Botanical drugs are defined as complex mixtures that lack a primary active ingredient that has been documented for prior substantial human use. These mixtures can be composed of vegetable materials, parts of plants, algae, macroscopic fungi, or combinations of them [16,17].

*Anacardium* genus (Anacardiaceae) consists of approximately 20 species, and *Anacardium microcarpum*, *Anacardium humile,* and *Anacardium occidentale* are the most widely studied, because of their medicinal and nutraceutical properties. Their common name is cashew, marañon, or cajú [18,19,20]. The leaves and flowers of *A. occidentale* have been used in traditional medicine to treat skin lesions and diarrhea, and as anti-inflammatory agents. Some formal studies for this species report antibacterial and antifungal activity and strong antioxidant action, attributed to the presence of anacardic acids, cardanol, gallic acid, catechin, and quercetin derivatives [21,22,23]. Regarding its antifungal potential, its activity has been briefly described before by a few authors against *Candida albicans* and *Candida tropicalis*, both reference strains [22,24,25].

The present study aimed to evaluate the anti-candida activity of the ethanol extract of *A. occidentale* leaf and determine its possible mechanism of action in *C. albicans*. Treatment with the extract induced an accumulation of intracellular ROS and mitochondrial dysfunction, and showed no cytotoxicity or hemolytic activity at the concentrations tested.

## 2. Materials and Methods

### 2.1. Strains, Culture and Chemicals

Ten Candida species were employed in this study, 4 wild types, viz. *C. albicans* ATCC SC5314, *C. albicans* PUJ/HUSI 256, *C*. *auris* PUJ/HUSI 435, and *C*. *auris* PUJ/HUSI 537, and six heterozygous haploid deletion *C. albicans* strains, purchased from the library developed by Xu, Deming et al. 2007 [26]. The clinical isolates were provided by San Ignacio University Hospital and the microorganism collection of Pontificia Javeriana University (Bogotá, Colombia). The mutant strains are detailed in Table 1. All strains were kept at −80 °C until reactivation. For reactivation, they were subcultured twice on yeast peptone dextrose (YPD) agar and incubated at 30 °C for 24–48 h. Amphotericin B (AmB), dimethyl sulfoxide (DMSO), 2′,7′-dichlorofluorescin diacetate (DCF-DA), rhodamine 123, *N*-acetyl-*L*-cysteine (NAC), fluconazole (FLC), sodium azide, FITC annexin V, and propidium iodide (PI) were purchased from MilliporeSigma (St. Louis, MO, USA) *A. occidentale* extract, AmB, and DCF-DA were dissolved in DMSO, rhodamine 123 in ethanol, and NAC, sodium azide annexin V, and PI in PBS, and frozen at −20 °C until use. In each assay, the content of DMSO was below 10%.

### 2.2. Plant Material and Extraction

Leaves of *Anacardium occidentale* were collected in April 2019, in the municipality of Orito, Putumayo, Colombia (0°3704500 N, 76°5105500 W). Biologist Néstor García from the herbarium of the Pontificia Universidad Javeriana carried out the taxonomic determination of the species, and a voucher was deposited with the collection number HPUJ-30548. The plant material was prewashed with 5% hypochlorite and water, dried in an oven with circulating air at 40 °C for 72 h, and then ground in a blade mill. The dried and ground plant material was extracted by percolation with EtOH 96% in a ratio of 1:10 (*w*/*v*), at room temperature, and protected from light in 4 cycles of 24 h each (with solvent changes). The extracts from the different cycles were pooled and concentrated under reduced pressure by rotary evaporation at a temperature of 40 °C. They were stored at room temperature in amber vials duly labelled for later analysis.

### 2.3. Chromatographic Analysis of Anacardium Occidentale Leaf Extract

#### 2.3.1. Preparation of Standard and Sample Solutions

A total of 2.5 mg of the dried extract was weighed and diluted in 1 mL of water/MeOH (LC-MS grade) solution (1:1, *v*/*v*). Standard solutions of phenolic compounds (quercetin, rutin, gallic acid, and chlorogenic acid) were prepared at 100 µg/mL with the same solvents. Standard solutions and sample solutions were filtered using a 0.22 μm filter.

#### 2.3.2. Ultra-Performance Liquid Chromatography Analysis

The chemical profile was obtained using UPLC-PDA on an Acquity H Class UPLC Waters^®^ (Milford, MA, USA) with a photodiode array detector (PDA), quaternary pump, degasser, column oven, and autosampler. The stationary phase was a Phenomenex^®^ Kinetex C18 (Torrance, CA, USA) (75 mm × 2.1 mm; 2.6 µm). The mobile phase was performed using 0.1% *v*/*v* formic acid (solvent A) and acetonitrile (solvent B). The gradient of the mobile phase was: 0–8 min, 13% B; 8–12 min, 13–20% B; 12–15 min, 20–22% B; 15–18 min, 22–27% B; 18–20 min, 27–30% B; 20–28 min, 30–35% B; 28–34 min, 35–90% B; 34–35 min, 90–13% B. The flow of the mobile phase was set at 0.4 mL/min. The UV–Vis spectrum was set between 200 and 500 nm, and the chromatograms were recorded at 350 nm. UHPLC-ESI-MS-QToF analysis was performed on a Nexera LCMS 9030 Shimadzu Scientific-Instruments^®^ (Columbia, MD, USA). The same chromatographic conditions were employed. The ionization method was ESI operated in negative ion mode. The capillary potentials were set at +3 kV, drying gas temperature 250 °C, and flow rate of drying gas 350 L/min.

### 2.4. Determination of Minimum Inhibitory Concentration

The minimum inhibitory concentration (MIC) values against *C. albicans* and *C. auris* were detected via the broth microdilution method, according to the Clinical and Laboratory Standards Institute (CLSI) guidelines (M27-A3) with some modifications [27]. ln a 96-well microplate, serial dilutions were made of the extract from 3.9 to 1000 µg/mL with RPMI-1640 medium (RPMI; Gibco, Grand Island, NY, USA) (100 µL), then 100 µL of the adjusted inoculum (1–5 × 10^3^ cells/mL) was added, the microplate was incubated at 37 °C for 48 h, and afterward a visual and spectrophotometric reading (600 nm) was performed. Controls were used as follows: Fluconazole (FLC) served as a positive control, growth control with no treatment, and Roswell Park Memorial Institute (RPMI) without inoculum as a sterile control. Additionally, the MIC values for FLC were known previously: *C. albicans* ATCC SC5314 (1 µg/mL), *C. albicans* PUJ/HUSI 256 (64 µg/mL), *C. auris* PUJ/HUSI 435 (8 µg/mL), and *C. auris* PUJ/HUSI 537 (128 µg/mL) [28]. The MIC was considered to be the minimum concentration with zero visible growth. Absorbance values were taken and considered to be the endpoint value. All assays were performed in triplicate.

### 2.5. Growth Kinetics Test

Concentrations of 15.6 to 250 μg/mL of the extract (150 μL) were evaluated in 100-well microplates (Honeycomb) using RPMI-1640 medium (RPMI; Gibco, Grand Island, NY, USA). Then, 150 μL of cell suspension (0.5–2.5 × 10^3^ cells/mL) was added, and the plates were incubated in a Thermo Labsystems Type FP-1100-C Bioscreen C (Piscataway, NJ, USA) at 30 °C for 48 h with constant agitation. The absorbance reading was performed every hour at 600 nm. The controls were the same as for the MIC test. All curves were performed in triplicate. The doubling time was calculated as reported by Murakami et al. [29]
$$Doubling\;time=\frac{\left[\ln\frac{2}{\ln\left(1+m\right)}\right]}{10}$$

### 2.6. Viability with Confocal Microscopy

Initially, a microdilution in broth was carried out, the same as for the determination of the MIC. After 48 h of incubation, a LIVE/DEAD™ Yeast Viability kit from Thermo Fisher Scientific (Waltham, MA, USA) was used and carried out according to the supplier’s instructions. Briefly, the multiwell plate was centrifuged for 5 min at 10,000 *g*, the supernatant was discarded, and 80 µL of PBS, 20 µL of FUN1™, and 100 µL of Calcofluor™ White M2R were added, incubated for 30 min at 30 °C, and taken to an Olympus FV 1000 confocal microscope (Tokyo, Japan). The wavelengths used were 405, 532, and 488 nm. This viability kit combines a two-color fluorescent probe for yeast viability, FUN1™, and a cell wall surface-binding fluorescent reagent, Calcofluor™ White M2R. In this way, if the integrity of the plasma membrane is preserved, the metabolic function of the yeast will be observed when it converts the yellow-green-fluorescent intracellular staining of FUN1™ into red-orange intravacuolar structures. Calcofluor White M2R will label chitin with blue fluorescence regardless of metabolic state [30].

### 2.7. Measurement of Intracellular ROS Production Assay

ROS levels were measured using 2′,7′-dichlorofluorescein diacetate (DCFH-DA). An inoculum of 1 × 10^8^ cells/mL was treated with different concentrations of extract for 2 h. Subsequently, two washes with PBS were carried out, and the cells were resuspended in 100 µL of PBS plus an aliquot of 20 µL of the indicator DCFH-DA (10 µg/mL) and incubated at 35 °C for 30 min. Fluorescence reading was performed with a Varioskan spectrofluorometer LUX (Thermo Scientific™, Waltham, MA, USA) (485 nm excitation wavelength: 535 nm emission wavelength). To determine the relationship between the generated ROS and cell death, a ROS scavenger was used: N-Acetyl-L-cysteine (NAC). For this, 3 × 10^8^ cells/mL were suspended in sodium acetate (12.5 mM) and incubated at 37 °C with NAC (60 mM) for 30–60 min [31,32].

### 2.8. Detection of Mitochondrial Function

The mitochondrial membrane potential is a critical point in the generation of ATP through the respiratory chain. Rhodamine 123 (MilliporeSigma, St. Louis, MO, USA) is a fluorochrome that is sensitive to membrane potential and is especially concentrated in the mitochondria. To measure the effect of *A. occidentale* extract, 3 × 10^8^ cells/mL were treated with or without different concentrations of extract and controls for 2 h. Subsequently, the solutions were centrifuged. The pellet was resuspended in rhodamine 123 (25 µM) and incubated at 30 °C for 30 min. After this time, the cells were washed three times with PBS, and the fluorescence intensity was measured on a Varioskan LUX spectrofluorometer (Thermo Scientific™, Waltham, MA, USA) (480 nm excitation wavelength; 530 nm emission wavelength). Cells without treatment were used as the negative control, and cells treated with 15 mM sodium azide were used as the positive control [31,33,34].

### 2.9. Scanning Electron Microscopy (SEM)

The yeast strains were pretreated with 2 MIC, MIC, and 0.5 MIC of extract in PBS for 24 h. Later, the solutions were washed twice with PBS, fixed with 100 µL glutaraldehyde 2.5%, and left at room temperature for 18 h after fixation. The suspensions were washed twice with 70% ethanol at 5 min intervals, three times with 95% ethanol at 10 min intervals, and three times with 100% ethanol at 20 min intervals. The samples were stored in 100% ethanol until analysis [35]. Then, the cells were observed under a Tescan Lyra 3 microscope (Brno-Kohoutovice, Czech Republic) at the microscopy center of the Universidad de Los Andes.

### 2.10. Transmission Electron Microscopy (TEM)

To visualize the effect of *A. occidentale* on the *C. albicans* organelles, TEM observation was performed. ATCC SC5314 cells were treated with 2 MIC, MIC, and 0.5 MIC of extract in PBS at 30 °C for 2 h. Cells without drug treatment served as the control. The cells were harvested by centrifugation at 1000 *g* for 5 min. The pellets were fixed, desiccated, and embedded, as previously described [31]. Then, the cells were observed under a Tescan Lyra 3 scanning electron microscope (Brno-Kohoutovice, Czech Republic).

### 2.11. Cell Death Assay with Propidium Iodine Staining

The cell death generated by the extract was measured by propidium iodide (PI) staining. PI is impermeable to cells with an intact plasma membrane; however, when cell integrity is compromised, it enters the nucleus, where it complexes with DNA, making the nucleus highly fluorescent [36]. Briefly, *C. albicans* ATCC SC5314 (1 × 10^7^ cells/mL) was treated with MIC and 2 MIC of *A. occidentale* extract for 4 h at 37 °C. After this time, the cells were washed with PBS, and 5 µL of calcofluor white (CW) (5 µg/mL) and PI (50 µg/mL) were added to the pellet, finishing with PBS to a final volume of 500 µL. Subsequently, the staining was left for 10 min at 4 °C in darkness and washed two times with PBS. The samples were observed under a LEICA DMi8 microscope (Leica Microsystems, Madrid, Spain).

### 2.12. Hemolytic and Cytotoxic Activity Assays

A suspension of red blood cells was prepared, and the hemolytic assay was performed following a published protocol [37]. Then, 190 µL of erythrocyte suspension was treated with 10 µL of the extract (3.9–500 µg/mL) in round-bottom 96-well plates and incubated at 37 °C for 2 h. Afterward, the plate was centrifuged at 500× *g* for 5 min to separate non-lysed erythrocytes. The supernatant liquid was transferred to a flat-bottom 96-well plate. Hemolysis was determined by measuring the absorbance of free hemoglobin in the solution at 450 nm. PBS was used as a negative control, while Tween 20 (20% *v*/*v*) in PBS was used as a positive control.

The cytotoxicity of the extract was evaluated on L929 mouse fibroblasts using a MTT colorimetric assay, as previously described [38]. 2.1 × 10^4^ cell fibroblasts were cultured in a flat 96-well plate and allowed to adhere and proliferate for 24 hours in Dulbecco’s Modified Eagle’s (DMEM) medium at 37 °C and 5% CO_2_. Subsequently, aliquots of *A. occidentale* (3.9–500 µg/mL) were added to wells containing the adhered cells. The resulting cell cultures were incubated for 24 h at 37 °C in a 5% CO_2_ atmosphere. After incubation, the medium was removed, the cells were washed with PBS, and 30 μL of MTT at a concentration of 1 mg/L in PBS was added. The cells were incubated again at 37 °C for 4 h, MTT was removed and 100 μL of DMSO was added to solubilize the formazan crystals. After 30 min of incubation at 37 °C, the absorbance at 575 nm was measured using a BioTek ELx800 absorbance reader (Santa Clara, CA, USA). Incomplete culture medium with 10% MTT was used as a negative control. In addition, cell viability at the half-maximum inhibitory concentration (IC_50_) was calculated by plotting viability versus log (concentration).

### 2.13. Statistical Analysis

The results of the experiments were tabulated in Excel and analyzed through analysis of variance (ANOVA), followed by Tukey’s test using GraphPad Prism 8 software. Values of *p* < 0.05 were considered significant.

## 3. Results and Discussion

### 3.1. Chromatographic Analysis of the Extract

To obtain a qualitative profile of the crude extract of *A. occidentale* leaves, an analytical chromatographic method was developed. The chromatogram of the extract is shown in Figure 1a,b. Several compounds with retention times between 10.0 and 35.0 min can be seen (Table 2). The most abundant compound in the extract (peak 10) was assigned as agathisflavone. This compound is a biflavonoid, obtained through the *C-C* coupling of two molecules of apigenin [39], and has been previously reported in *A. occidentale* [40]. Biological activity has been reported for this compound, such as antitumor activity and antibacterial activity [41,42]. Moreover, this flavone could be a marker for *A. occidentale* extracts.

Previous studies have described the presence of flavonoids and tannins in some organs of plants from the Anarcadiaceae family [43,46]. Flavonoids are usually the major secondary metabolites, and they play a significant role in the environmental response. Based on Table 2, different compounds were tentatively identified based on their high-resolution mass, UV spectra, retention time, and comparison with the data in the literature. Among the compounds identified, glycosylated flavonoids derived from quercetin nuclei were predominant. Konan and coworkers [47] developed a chemical analysis of a methanolic extract from *A. occidentale* leaves. The composition of the methanolic extract was determined to be glycosylated quercetin derivatives and other phenolic compounds.

### 3.2. Antifungal Activity of A. occidentale Leaf Extract against Candida *spp.*

To determine the susceptibility of *Candida* spp. to the extract, the broth microdilution method, based on CLSI standard M27-A3, was performed against both the reference strain and clinical isolates of Candida strains. The results showed that *A. occidentale* exhibited an antifungal effect. The MIC value was 62.5 µg/mL for *C. albicans* ATCC SC5314, and for fluconazole (FLC)-resistant strain *C. albicans* PUJ/HUSI 256 (Figure 2a). To determine the maximum fungicide concentration (MFC), aliquots of concentrations with no visible growth were subcultured on an agar plate and incubated for 48 h. The MFC was 1000 µg/mL for all of them. At this concentration, the cells were unable to recover their growth, while at concentrations between 125 and 500 µg/mL, the cells were still able to grow on the agar plate after 48 h of incubation (Figure 2b).

Little information has been detailed with respect to the antifungal activity of *A. occidentale*. Some studies have shown the MIC ranges between 400 and 2000 μg/mL against *C. albicans* [25,44,48]. Other research has reported no activity against *C. albicans* [22]. Additionally, a study on A. humile, with phytochemical characteristics very similar to those of *A. occidentale*, reported an MIC of 400 μg/mL [18]. Accordingly, the *A. occidentale* leaf extract from this study exhibited MICs reduced by a factor of 8, which is relevant, since they are the lowest concentrations reported thus far. This activity can be attributed to the presence of flavonoids, flavonols, and glycosylated flavonoids derived from quercetin, identified by means of UPLC and high-performance liquid chromatography (HPLC) contained in the ethanolic extract (Table 2) (Figure 1), since their role as an antifungal has already been reported [45,49,50].

Regarding the cut-off points, there is no established classification scale that makes it possible to determine the level of activity in an extract. However, on comparing previous studies under the same experimental conditions [22,25,44], it was established that the ethanolic extract of *A. occidentale* exhibits significant activity, as described previously. With respect to solvents, it was previously shown that dichloromethane and the methanol extracts of *Punica granatum*, *Syzygium cumini*, *Rosmarinus officinalis*, *Arrabidaea chica*, *Mentha piperita*, and *Tabebuia avellanedae* exhibited great activity, with MIC values ranging from 1 to 60 μg/mL against various *Candida* strains (*C. albicans*, *C. dubliniensis*, *C. parapsilosis*, *C. tropicalis*, *C. guilliermondii*, *C. utilis*, *C. krusei*, *C. lusitaniae*, *C. glabrata*, *C. rugosa*). On the other hand, the antifungal potential of the hydroethanolic extract of Astronium urundeuva leaves was evaluated against *C. albicans* and *C. glabrata*, demonstrating significant activity, with MIC_90_ values ranging from 0.24 to 15.62 μg/mL, thereby confirming our findings [51,52].

### 3.3. Effect of A. occidentale on the Growth Curve of Candida Species 

To determine the effect of the extract on proliferation, growth kinetics curves were generated. The extract of *A. occidentale* affected the growth kinetics of *C. albicans*, increasing the doubling time (T_d_). As shown in Figure 3, at 62.5 μg/mL, *C. albicans* ATCC SC5314 had a T_d_ of 7.6 h compared to the untreated control, 2.8 h (*p* < 0.0001), suggesting that the extract partially inhibited its proliferation (below 50%) after 48 h, while at 250 μg/mL, no growth was detected. However, as carried out in the MIC determination assay, the cells were subcultured and recovered their growth; therefore, it cannot be affirmed that there was a fungicidal effect at that concentration. Interestingly, despite exhibiting the same MIC, at 62.5 μg/mL, the extract more strongly inhibited the growth of the FLC-resistant clinical isolate *C. albicans* PUJ/HUSI 256, with a Td of 17.41 h, compared to the sensitive strain *C. albicans* ATCC SC5314, with 7.6 h. The T_d_ for every concentration is shown Table 3.

Since the invasive fungal disease caused by *C. albicans* can start from an endogenous source, such as the mucous membranes of immunosuppressed individuals [49], this result is important, because the extract can significantly slow the reproduction and proliferation of the yeast, which would prevent the onset of the disease in its early stages [50,51,52]. In the early stage, the most widely used therapy is antifungal prophylaxis, which prioritizes administration in patients with risk factors or a high susceptibility to colonization by *C. albicans* before the appearance of signs and symptoms of the disease [50,51,53]. The commonly used agent is fluconazole; meanwhile, resistance rates are frequent [2].

### 3.4. Confocal Microscopy

To corroborate the results obtained from the broth microdilution and the kinetic curves, the LIVE/DEAD™ Yeast Viability kit from Thermo Fisher Scientific was used and visualized by confocal laser scanning microscopy (CLSM). 0.5 MIC, MIC, and 2 MIC were employed against *C. albicans* ATCC SC5314. Viable yeasts use the green fluorescent dye FUN1 and incorporate it into red-orange intravacuolar structures, indicating plasma membrane integrity and the metabolic activity of the yeast [54]. Calcofluor White M2R tags chitin with blue fluorescence, regardless of the metabolic state. A control with no treatment and a dead yeast cell control treated with hypochlorite were used. As shown in Figure 4, in the bright field, a decrease in the number of cells was very visible when treated with MIC and 2 MIC. In addition, there is a notable reduction in the red-orange.

### 3.5. Scanning and Transmission Electron Microscopy

To examine the cell morphology of *C. albicans*, scanning electron microscopy (SEM) was first performed. After exposing the yeasts to *A. occidentale* for 48 h at MIC and 2 MIC, masses of debris and intracellular leakage were observed on the cell surface in *C. albicans* ATCC SC5314 (Figure 5d–f) that intensified as the concentration increased (Figure 5g–i). There were no evident changes in the characteristic oval structure or the presence of pores, wrinkles, or other abnormal morphologies.

Second, scanning transmission electron microscopy (STEM) was utilized to identify changes in the ultrastructure of *C. albicans* ATCC SC5314 in response to treatment. Untreated *C. albicans* cells were intact, uniform, and oval-shaped (Figure 6a–d). After treatment with 2 MIC (125 μg/mL) *A. occidentale* for 4 h, the *C. albicans* cells became highly irregular, and some vacuoles around the nucleus and cytoplasm (Figure 6e,g,h), some cellular membrane retraction (Figure 6e,g), and irregular edges on the cell wall (Figure 6e–h) could be seen. This is the first investigation where these microscopies were performed, and the loss of intracellular content was demonstrated (Figure 5). TEM showed irregularities at the cell borders, as well as membrane retraction (Figure 6e–h), which may explain the leakage of intracellular content, suggesting that the *A. occidentale* leaf extract would be causing disturbances in the ergosterol and sphingolipid levels [55,56] responsible for the porosity, fluidity and ultrastructure of the plasma membrane and cell wall. Studies report that under stress conditions, there is a downregulation of the ERG1, ERG3, and ERG11 genes associated with ergosterol synthesis [56,57], and of the IPT1 gene associated with sphingolipid biosynthesis [58]. To understand more about this mechanism, more experiments must be performed.

### 3.6. Effect of A. occidentale on Intracellular ROS Accumulation

In none of the works where the antifungal activity was reported, either by microdilution in broth or by diffusion in disk, was an approach to the mechanism of action carried out. To better understand how *A. occidentale* works, first the role of the extract in oxidative damage was evaluated. The *A. occidentale* extract was tested at 0.5 MIC, MIC, and 2 MIC. Amphotericin B (AmB) was used as a positive control for oxidative damage, and N-acetyl glucosamine (NAC) was added as a ROS scavenger (Figure 7). It is generally known that exacerbated ROS production is one of the most sensitive characteristics associated with the universal mechanism of action of drugs. In antifungals, AmB exerts irreversible oxidative damage, contributing to its fungicidal activity [59]. Therefore, the effect of *A. occidentale* on the intracellular ROS level was determined, and this demonstrated that the treatment induced ROS accumulation in a dose-dependent manner compared to the untreated control (*p* < 0.05) (Figure 7). When NAC was added, the stress induced by both AmB and extract concentrations was relieved, demonstrating that the fluorescence obtained is due to the accumulation of intracellular ROS.

Furthermore, endoplasmic reticulum stress is caused by an accumulation of misfolded protein. This process contributes greatly to the formation of intracellular ROS due to the formation of disulfide bonds during folding [60,61,62,63]. To determine hypersensitivity or resistance phenotypes and provide experimental data on the possible mechanism of action of the extract, strains heterozygous for genes involved in the unfolded protein response (UPR) process in endoplasmic reticulum stress were exposed to the extract. ReviThis approach is a new tool in the search for new antifungal candidates [26,64,65]. As shown in Figure 8a, there were no notable differences compared to the wild type ATCC SC5314. However, in the spot assay (Figure 8b), a slight reduction in spot density was observed, which was slightly more notable in the Δhac1, Δkar2 and Δhog1 mutants. The three strains identified after exposure to the extract represent different aspects of its possible mechanism of action: Hog1p, a major regulator that is probably involved in upstream signaling from the cytoplasm; and the two UPR-associated proteins, the transcription factor (Hac1p), which dislocates to the nucleus and activates the expression of UPR genes, such as those encoding endoplasmic reticulum-resident chaperones (Kar2p) mainly responsible for protein folding and stress relief; an interaction between the extract and the three strains mentioned above was evidenced, to better understand the possible interaction, see Figure 9 (Table 1) [63,66,67].

On the other hand, it has been described that the mutant of the UPR-triggering receptor Ire1p, located in the lumen of the endoplasmic reticulum, exhibited higher sensitivity to stressors [66,67]. In the present study, no growth defects of mutant Ire1p were observed. Additionally, the effect on Δhog1 suggests that the mechanism of action is multifactorial, since it has been described that there is a triad between metabolic respiration, the UPR signaling mediated by Ire1p and the response to osmotic and cell wall stress mediated by Hog1p [68], which was corroborated by the damage to the ultrastructure of the cell wall and plasma membrane observed by TEM (Figure 6e–h).

### 3.7. Effect of A. occidentale on the Mitochondrial Potential (mtΔψ) 

As previously described, ROS are a byproduct of the cellular metabolism generated primarily by oxidative phosphorylation and protein folding, processes that occur in mitochondria and the endoplasmic reticulum, respectively [69,70,71]. The increase in intracellular ROS strongly affects the function of mitochondria. In this organelle, the steps prior to cell death occur [70,72,73]. Therefore, the effect of the extract on the loss of membrane potential was evaluated using rhodamine 123. As shown in Figure 10, after treatment with MIC and 2 MIC, the fluorescence significantly increased compared to the positive control and the positive control with sodium azide, *p* < 0.007 and *p* < 0.002, respectively.

The extract induced the loss of mtΔψ, leading to mitochondrial dysfunction, which was alleviated when NAC was used as an antioxidant agent (Figure 10). Sodium azide is an inhibitor of complex IV of the mitochondrial respiratory chain, also called cytochrome c oxidase; its function is vital for the generation of ATP [71,74]. Taken together, these results suggest that at MIC and 2 MIC concentrations, *A. occidentale* extract induced an increase in intracellular ROS that led to the loss of mitochondrial function, impairing the metabolic process and possibly decreasing the generation of ATP, which is vital for anabolic processes such as cell reproduction [75]. In this investigation, it was also correlated with delayed doubling times (Figure 3 and Table 2) and the inability to metabolize FUN1 (Figure 4). However, another assay for oxygen consumption and the measurement of intracellular ATP must be performed.

### 3.8. Cell Death Assay with Propidium Iodine

To investigate the potential effect of *A. occidentale* extract on cell death, *C. albicans* ATCC SC5314 was subjected to different concentrations of the extract, specifically MIC and 2 MIC. Cell death was assessed by staining the cells with propidium iodide (PI), which identifies membrane-permeabilized cells, while Calcofluor White was used to stain the fungal cells. Our findings revealed the minimal tagging of yeast and hyphal components with PI in both strains upon treatment with *A. occidentale* extract. However, the results depicted in Figure 11 suggest that the extract at a concentration of 125 μg/mL somehow disrupts the plasma membrane of ATCC SC5314 cells, facilitating the entry of the fluorescent dye. Notably, the detection of clotrimazole signals exhibited distinctive red fluorescence localized within the intracellular compartments of the cells. Regarding membrane integrity, *C. albicans* slightly incorporated PI inside the cell after treatment with *A. occidentale*. These results could be further supported by the drop in cell viability, indicating the extract’s potential fungistatic effect. In contrast, as reported, most polyene antimycotics have been attributed to their ability to alter the yeast membrane permeability by their fungicidal action related to direct membrane damage [76], suggesting that *A. occidentale* could have a different mechanism of killing in eukaryotic target cells, or that it has some intracellular targets as well [77].

### 3.9. Toxicity of A. occidentale Extract

The hemolytic activity was tested using a suspension of erythrocytes. The extract showed a percentage of hemolysis of 25% at the highest concentration. However, at 250 μg/mL, the levels were minimal, including the concentrations of the MIC and 2 MIC shown in Figure 12a. Second, cytotoxicity was evaluated by treating L929 fibroblast with the same concentrations of the extract. According to Figure 12b, at 62.5 μg/mL of extract, the viability percentage was 70%, which was maintained up to 500 μg/mL. Taken altogether, the extract showed low toxicity.

Recently, Chaves de Araújo et al. [78] supported our findings, demonstrating that the *A. occidentale* extract exhibited minimal hemolytic activity at concentrations of 1, 10, 100, and 1000 μg/mL. In their research, this was attributed to the presence of tannins, since the biological functionality of the blood cells remained intact, maintaining cell membrane integrity. The L929 cells exhibited low cytotoxicity with the extract; this result is in accordance with Rodrigues Costa et al. [45] and Shabeeba M Ashraf and Krishnan Rathinasamy [79], whose reports confirmed that the extract did not affect the cell viability independently of the dose. Therefore, we can infer that the observed low cytotoxicity may be due to the bioactive compounds present in the *A. occidentale* extract, but this does not fully elucidate the mechanism. Studies have reported how flavonoids act as antioxidants and protect various cell types from stress-mediated cell injury [80,81]. This supports our results, since the extract of the present study is mainly composed of flavonoids.

### 3.10. Effect of A. occidentale on the Growth of Candida auris

After having obtained some promising results for *C. albicans*, it was decided to evaluate the effect of the *A. occidentale* extract on two strains of *C. auris*: *C. auris* PUJ/HUSI 435 sensitive to FLC (MIC: 8 µg/mL), and *C. auris* PUJ/HUSI 537 resistant to FLC (MIC: 128 µg/mL) and to AmB (MIC > 32 µg/mL) [6]. In the present study, it was shown that the inhibitory activity against *C. albicans* was the same as that against *C. auris*, since the MIC was 62.5 µg/mL. Interestingly, despite exhibiting the same MIC, the extract more strongly inhibited the growth of the multidrug resistant clinical isolate *C. auris* PUJ/HUSI 537, with a Td of 22.6 h compared to the antifungal-sensitive *C. auris* PUJ/HUSI 435, with a Td of 7.3 h (Figure 13a,b). In the same way, as was observed for *C. albicans*, the extract at MIC and 2 MIC exerts a fungistatic activity that is corroborated by the gradual loss of the ability to metabolize the FUN1 dye in confocal microscopy (Figure 13c), and the appearance on the surface of lumps that resemble the loss of some intracellular content without the appearance of physical alterations, such as pores or wrinkling, observed through SEM (Figure 13d). Regarding PI staining, it was observed that *C. auris* slightly incorporated PI inside the cell after treatment with *A. occidentale* (Figure 13e).

Only one experimental investigation with total plant extract has been reported to date for *C. auris*. Ashmawy et al. [82] showed the inhibitory activity of a compound isolated from the methanolic extract of Tephrosia apollinea at 200 µg/mL; on the other hand, the total extract only inhibited between 15% and 20% at a concentration of 900 µg/mL, indicating that *A. occidentale* exhibited significant promising activity, considering the public health concerns about this emerging pathogen highly tolerant to FLC and AmB. A recent study on the incidence of *C. auris* in patients colonized between 2020 and 2021 showed that 100% of the isolates were resistant to FLC and 50% to AmB in the first case episode [83].

## 4. Conclusions

The emergence of resistance to conventional drugs and the toxicity of high doses highlights the importance of developing new alternatives for treating invasive fungal diseases. NPs, mainly those of botanical origin, belong to a field that has recently been approved by the FDA, although to date there is no botanical drug or derivative thereof for use as an antifungal. *A. occidentale* has been used in traditional preparations for the treatment of diseases. In the present study, an ethanolic extract of leaves from *A. occidentale,* containing several glycosylflavonoids and with the biflavone agathisflavone as the major compound, was found to inhibit the growth and proliferation of *C. albicans* and *C. auris* at 62.5 µg/mL and 125 µg/mL. These results were corroborated by confocal microscopy, SEM, and TEM. Additionally, the extract induced the accumulation of intracellular ROS and mitochondrial dysfunction and did not show cytotoxicity or hemolytic activity at the concentrations tested. This is the first time that the mechanism of action of the plant has been explored, and it was shown that the flavonoids present in it may be related to this activity. This investigation expands the current knowledge of botanical drugs as a potential alternative to combat invasive fungal disease.

## Figures and Tables

**Figure 1 jof-10-00464-f001:**
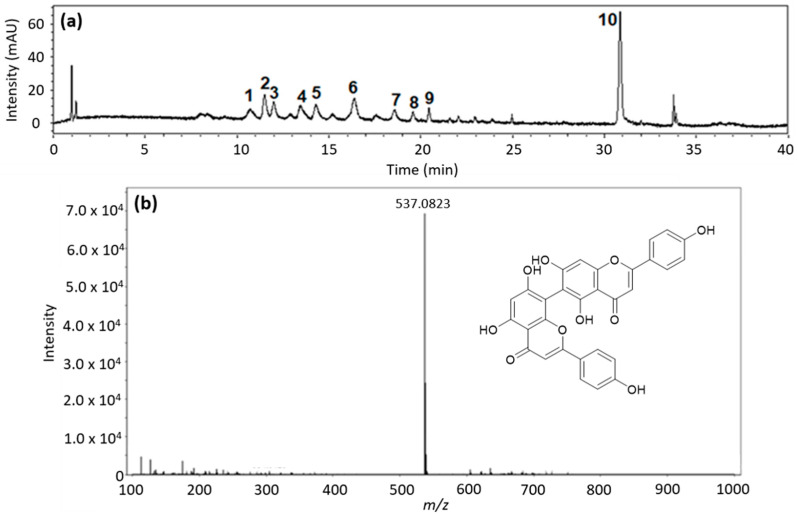
(**a**) Ultra-performance liquid chromatography (UPLC) chromatogram obtained from *A. occidentale* leaf extract at 350 nm. (**b**) Mass spectrum of agathisflavone.

**Figure 2 jof-10-00464-f002:**
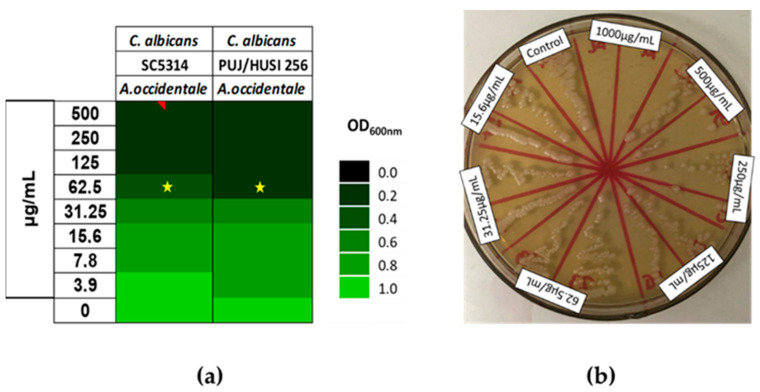
Minimal Inhibitory Concentration (MIC) and Maximal Fungicide Concentration (MFC) of *A. occidentale* against *Candida albicans* strains. (**a**) MIC by broth microdilution based on CLSI M27-A3. (**b**) MCF against ATCC SC5314 by subculturing on agar plate after MIC method. The star indicates the MIC value.

**Figure 3 jof-10-00464-f003:**
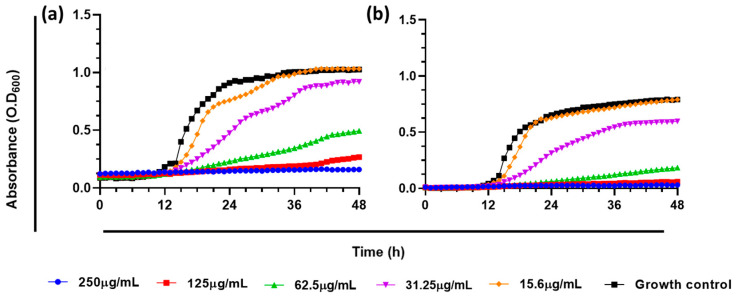
Effect of *A. occidentale* extract on growth kinetic of *C. albicans* strains. (**a**) *C. albicans* ATCC SC5314; (**b**) *C. albicans* PUJ/HUSI 256. The figure was constructed with information from three independent 48 h curves performed under the same protocol.

**Figure 4 jof-10-00464-f004:**
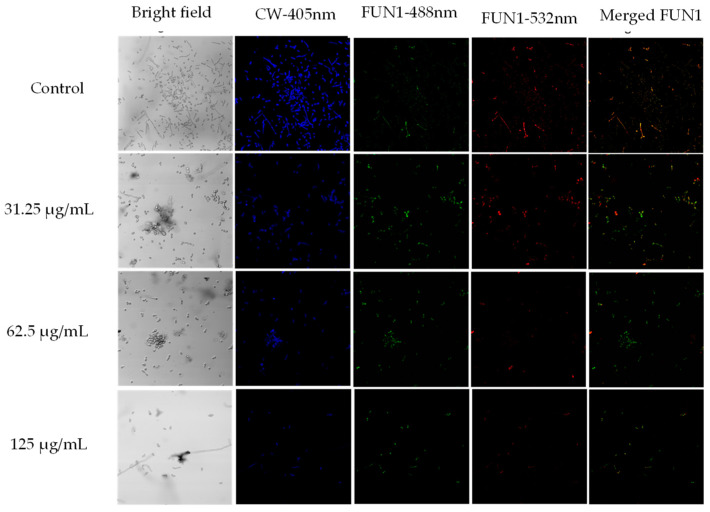
Confocal scanning fluorescence images of *C*. *albicans* ATCC SC5314 cells after exposure to 0.5 MIC (31.25 μg/mL), MIC (62.5 μg/mL), and 2 MIC (125 μg/mL) of *A. occidentale*. Live or dead intact cells stained with Calcofluor White M2R dye (blue fluorescence), live or dead cells stained with FUN1 488 nm (green fluorescence), active metabolic cells stained with FUN1 532 nm (red fluorescence), and a merged image, respectively. Untreated cells in PBS were used as a negative control. All images were taken with a 40× objective.

**Figure 5 jof-10-00464-f005:**
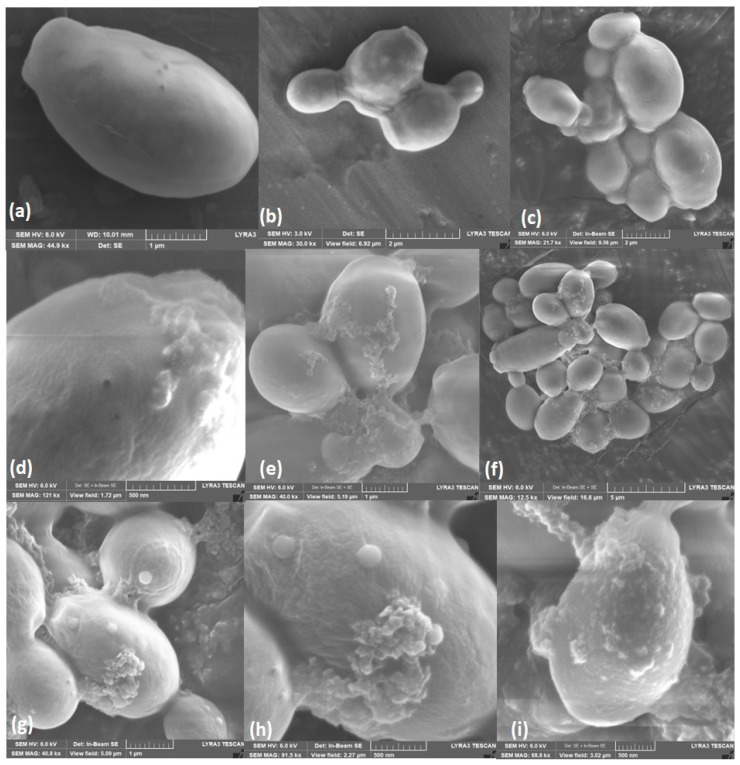
Scanning electron microscopy of *C. albicans* ATCC SC5314 cells treated with *A. occidentale* leaf extract. Approximately 1 × 10^8^ cells were incubated without (**a**–**c**, controls) or with 62.5 μg/mL (**d**–**f**) and 125 μg/mL of extract for 24 h. Masses of debris and cell leakage were observed on the cell surface after treatment. Image size: (**a**,**e**,**g**): 1 µm; (**b**,**c**): 2 µm; (**f**): 5 µm; (**d**,**h**,**i**): 500 nm.

**Figure 6 jof-10-00464-f006:**
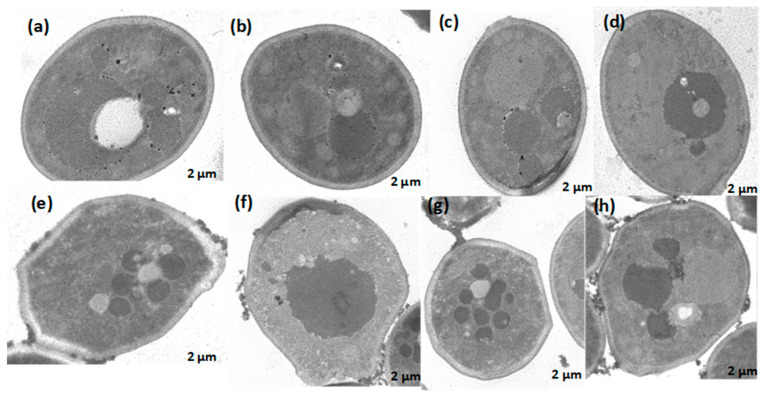
Transmission electron microscopy of *Candida albicans* ATCC SC5314 cells treated with *A. occidentale* leaf extract. Approximately 1 × 10^8^ cells were incubated without (panels **a**–**d**) or with 125 μg/mL extract (panels **e**–**h**) for 4 h. Still intact cells treated presented cytoplasmic changes with the presence of several microvacuoles (panels **e**,**g**,**h**), membrane retraction (**e**,**g**), irregular cell wall (panels **e**–**h**) and disintegrated nucleus envelope (**f**). Bar: ~2 μm.

**Figure 7 jof-10-00464-f007:**
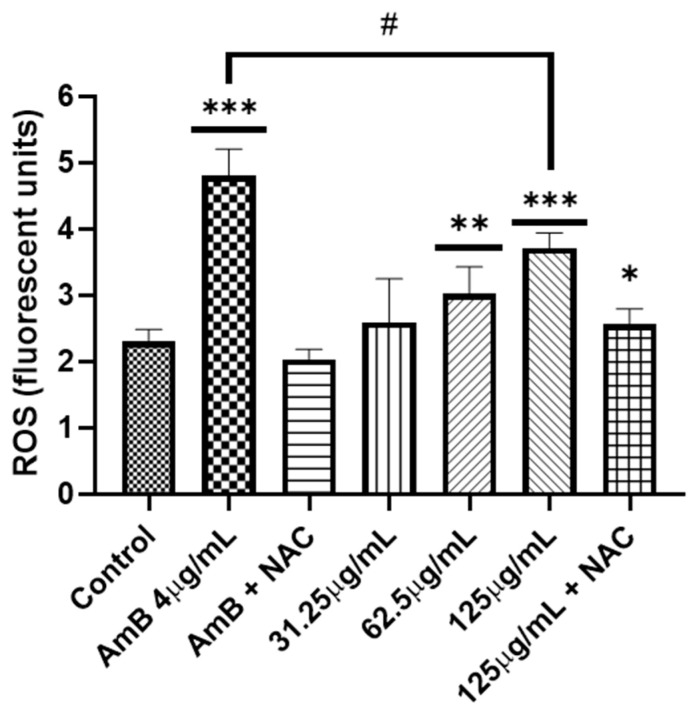
Measurement of ROS in *C. albicans*. Two concentrations (62.5–125 μg/mL) of *A. occidentale* were used; AmB 4 μg/mL was used as a positive control, and NAC 60 mM as ROS a scavenger. All data are represented with the mean ± SD. Fluorescence was detected with a DCF-DA. This figure was constructed with information from three independent 2 h experiments performed under the same protocol. Statistical analysis was conducted through ANOVA.. Significance was denoted by *p*-values as follows: (*) *p* < 0.005; (**) *p* < 0.005; (***) *p* < 0.001; (#) *p* < 0.01.

**Figure 8 jof-10-00464-f008:**
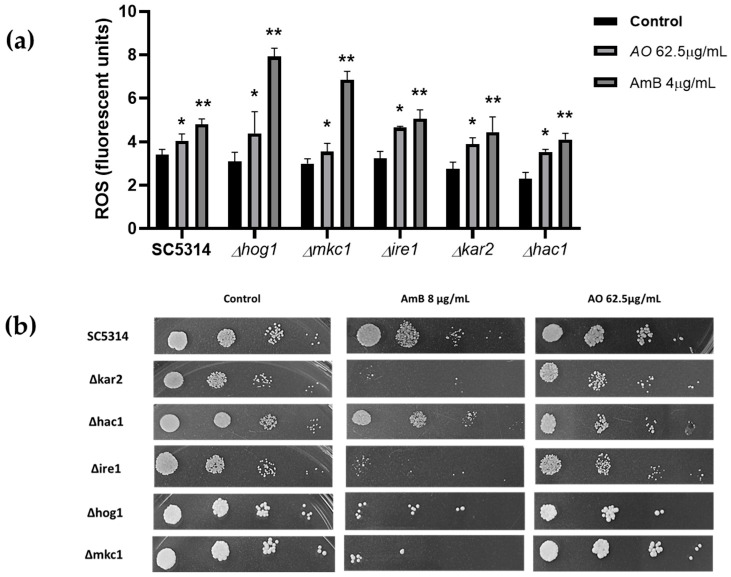
(**a**) Measurement of ROS in UPR mutants strain. We employed *A. occidentale* 62.5 μg/mL; AmB 4 μg/mL was used as a positive control. All data are represented with the mean ± SD. Fluorescence was detected with a DCF-DA. This figure was constructed with information from three independent 2 h experiments performed under the same protocol. (**b**) Phenotypic analysis of *C. albicans* UPR mutants. (*) *p* < 0.005; (**) *p* < 0.005.

**Figure 9 jof-10-00464-f009:**
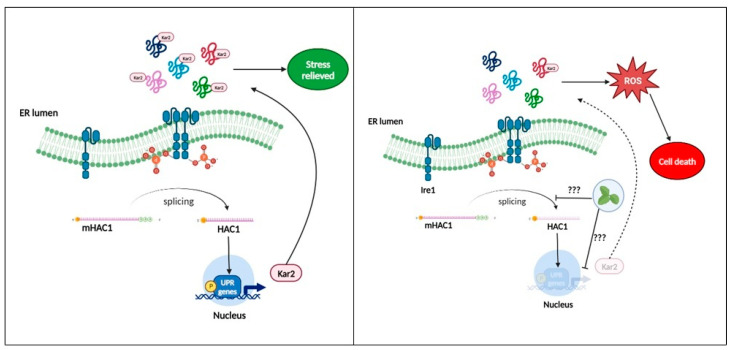
Possible mechanism of action associated with endoplasmic reticulum stress. (**Left**) Ire1p is activated by ER stress. Upon activation, Ire1p undergoes autophosphorylation and dimerization. HAC1 mRNA is spliced by activated Ire1p, then, transcription factor Hac1 upregulates UPR target genes, mainly Kar2p, to restore homoeostasis. (**Right**) In heterozygous strains,the Hac1p transcription factor is in low quantity and/or its translocation to the nucleus is affected, preventing efficient expression of the UPR target genes and, consequently, Kar2p cannot alleviate stress, contributing to the increase in ROS in the ER and leading to cell death by inefficient UPR.

**Figure 10 jof-10-00464-f010:**
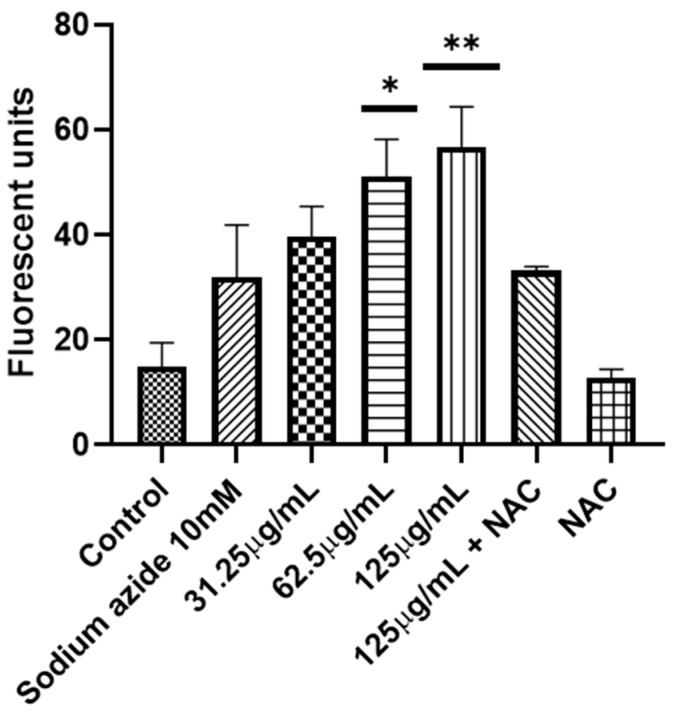
Effect on the mitochondrial function in *C. albicans*. Three concentrations of *A. occidentale* were used (31.25–125 μg/mL); sodium azide 10 mM was used as a positive control and NAC 60 mM as a ROS scavenger. All data are represented with the mean ± SD. Fluorescence was detected with rhodamine 123. The figure was constructed with information from three independent 2 h experiments performed under the same protocol. Statistical analysis was conducted through ANOVA. Significance was denoted by *p* values as follows: (*), *p* < 0.002; (**), *p* < 0.007.

**Figure 11 jof-10-00464-f011:**
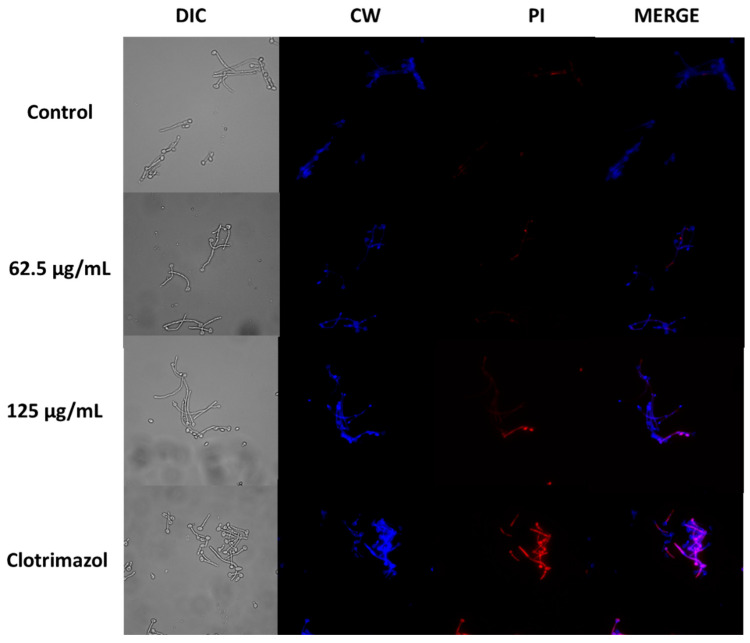
Confocal scanning fluorescence images of *A. occidentale*-induced inhibited growth in *Candida albicans* ATCC SC5314. Cells treated with 62.5 and 125 μg/mL of ethanolic extract after 4 h were stained with Calcofluor White (CW)/propidium iodide (PI). CW, excitation at 375 nm; PI, excitation at 555 nm. Scale bars, 10 μm.

**Figure 12 jof-10-00464-f012:**
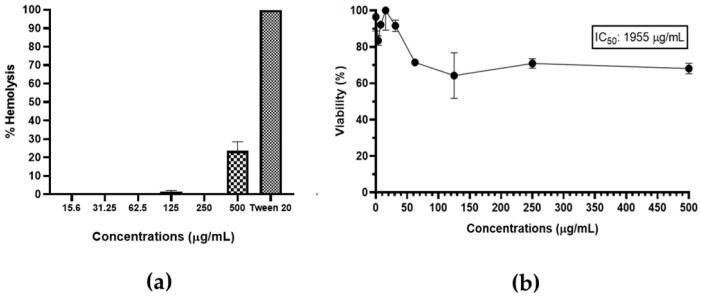
(**a**) Hemolytic profiles; Tween 20 was used as positive control; and (**b**) cell viability of fibroblasts L929 in the presence of *A. occidentale* extract.

**Figure 13 jof-10-00464-f013:**
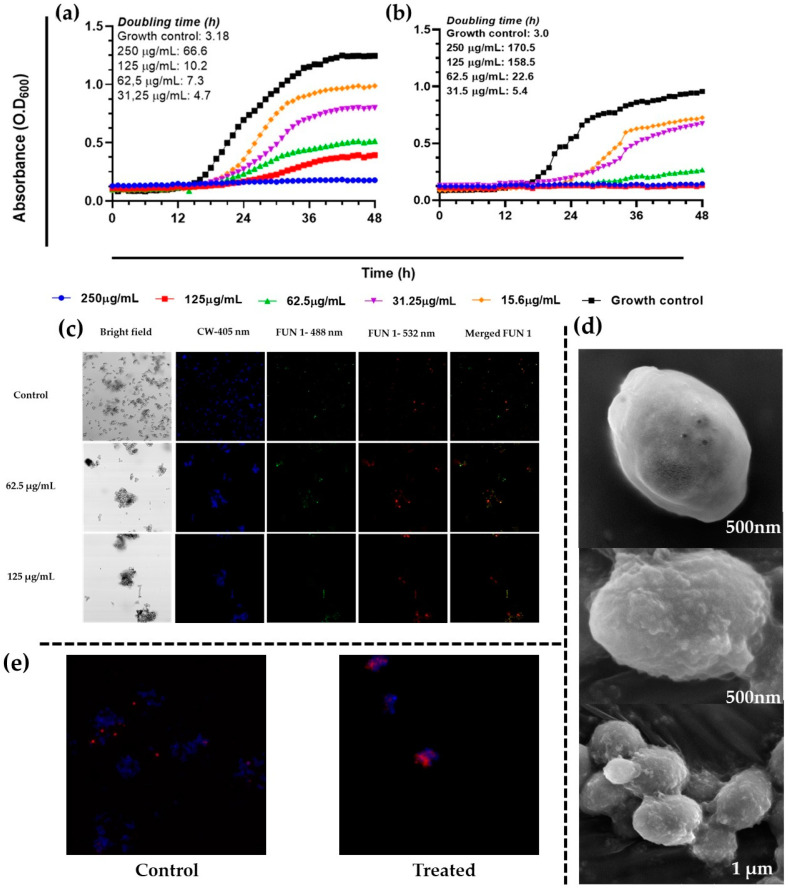
Effect of *A. occidentale* extract on the growth kinetic of (**a**) *C. auris* PUJ/HUSI *435* and (**b**) *C. auris* PUJ/HUSI 537. This figure was constructed with information from three independent 48 h curves performed under the same protocol; (**c**) images of *C*. *auris* PUJ/HUSI *435* after exposure to MIC and 2 MIC from the LIVE/DEAD™ Yeast Viability kit from Thermo Fisher via confocal laser scanning microscopy; 40× magnification; (**d**) SEM of *C. auris* PUJ/HUSI 435. The first photo is a cell without treatment, the second and third photos are cells treated with MIC, and then in (**e**) the permeability of the membrane was evaluated by means of PI staining after exposing *C. auris* PUJ/HUSI 435 for 4 h to MIC (images captured at 40× magnification).

**Table 1 jof-10-00464-t001:** List of heterozygous haploid deletion strains used in this study.

Strains *	Gene Ontology (GO) **
*C. albicans* Δhog1/HOG1	MAP kinase of osmotic-, heavy metal-, and core stress response; role in regulation of response to stress
*C. albicans* Δmkc1/MKC1	MAP kinase; role in membrane perturbation, or cell wall stress
*C. albicans* Δirei1/IREI1	Protein kinase involved in regulation of unfolded protein response
*C. albicans* Δkar2/KAR2	Chaperone with role in translocation of proteins into the endoplasmic reticulum
*C. albicans* Δhac1/HAC1	bZIP transcription factor with role in unfolded protein response
*C. albicans* Δero1/ERO1	Role in formation of disulfide bonds in the endoplasmic reticulum

* Source: [26]; ** http://www.candidagenomedatabase.org (accessed on 23 June 2024).

**Table 2 jof-10-00464-t002:** Peak assignments of the *A. occidentale* extract using UPLC-PDA/qTOF-MS in negative mode.

Peak No.	Compounds	Rt (min)	[M–H]^-^ *(m/z)*	λ_max_ (nm)	References
**1**	5-Methylcyanidin-3-*O*-hexoside	10.7	463.0762	282.1–514.2	[43]
**2**	Quercetin 3-*O*-α-L-rhamnoside	11.5	447.0922	255.9–349.5	[44]
**3**	Quercetin galloyl-*O*-deoxy-hexoside	12.0	599.1026	257.1–349.5	[45]
**4**	Quercetin 3-*O*-xylopyranoside	13.5	431.0954	257.1–349.5	[40]
**5**	Unknown flavonoid *	14.3	Nd	263.0–349.5	
**6**	Unknown flavonoid *	16.4	Nd	258.2–347.2	
**7**	Kaempferol 3-*O*-α-glucoside	18.5	433.0925	266.6–348.3	[40]
**10**	Agathisflavone	30.8	537.0823	271.3–334.5	[39,45]

**Table 3 jof-10-00464-t003:** Doubling times of *A. occidentale* against *Candida albicans*.

Treatment (μg/mL)	Doubling Time (Hours)	
	*C. albicans* ATCC SC5314	*C. albicans* PUJ/HUSI 256
**0**	2.78	3.43
**250**	84.05	153.4
**125**	22.58	55.61
**62.5**	7.61	17.40
**31.25**	3.16	4.31

## Data Availability

The original contributions presented in the study are included in the article, further inquiries can be directed to the corresponding author.

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
