# Peer review of "Unmasking the Antifungal Activity of Anacardium occidentale Leaf Extract against Candida albicans"

_jof, 2024, doi:10.3390/jof10070464_

Round 1

Reviewer 1 Report

1. The study evaluated the antifungal effects of  Anacardium occidentale  leaf ethanol extract against Candida albicans and C. auris, and it found that the extract caused changes in the morphology and structure of Candida albicans, as well as the induction of intracellular ROS accumulation.Nonetheless, despite the preliminary identification of the extract's main components as flavonoids, particularly methanolic extracts containing glycosylated quercetin derivatives and other phenolic compounds, the authors did not actually isolate and purify the specific active compounds for their experiments. All biological assays were conducted using a crude mixture. Given the abundance of prior research on the antifungal properties of cashew leaf extracts, the study does not present a substantial leap in terms of novelty.

2. While a slight decrease in spot density was observed for three mutants in the spot assay, there was no noticeable difference in ROS levels between the six mutated strains and the wild type after exposure to the extract in liquid culture conditions. The authors made extensive speculations to suggest a connection between the extract's antifungal activity and endoplasmic reticulum (ER) stress in fungi, but these remained mere conjectures, as they had not experimentally validated this hypothesis. To conclusively establish that the extract induces ER stress and mitochondrial dysfunction, additional experiments would be needed to substantiate the claim.

3.Line 380-383  The author inferred "This damage is caused by disruptions in ergosterol and sphingolipid levels" solely based on transmission electron microscopy results, which would require further experimentation for validation.

4.In the ROS (Reactive Oxygen Species) experiment, the author chose amphotericin B as a positive control. Amphotericin B is a commonly used antifungal drug in clinical practice, and its bactericidal activity is significantly stronger than fluconazole. The MIC against Candida albicans is generally 0.25μg/mL, However, the author used a concentration of 4 μg/mL in their experiment. Why was such a high concentration chosen? Furthermore, amphotericin B primarily targets fungal cell membranes by binding to ergosterol, thereby disrupting membrane integrity for its fungicidal effect. Is it appropriate to use it as a positive control for oxidative damage? Additionally, in the spot assay (Figure 8b), amphotericin B was employed at a concentration of 8 μg/mL, which is far beyond the growth threshold for Candida. How can SC5314 still be observed to grow in Figure 8?

1. Were the six heterozygous haploid deletion strains used in the article synthesized in the laboratory or procured from a company? Please provide a clear statement. Furthermore, the text should include an explanation for the selection of these particular strains

2. Line 152  RPMI medium should be clearly indicated as RPMI 1640 medium. Is there a 100 well plate? The author is requested to confirm this detail.

3. In the MIC determination experiment, the author employed a positive control value that was derived from a previously known value. Where does this value come from? Reference or results from previous laboratory measurements? However, in reality, due to the different sources of reagents and operators used in the antibacterial activity experiment, positive controls need to be set every time, which is also a quality control requirement. Therefore, it is inappropriate for the author to use the previous data in this article.

4.The SEM experiment observed bacterial morphology after 24 hours of extract treatment, while TEM observed structures after only 2 hours. Why is there such a significant time difference in the setup?

5.Which type of red blood cells were used in the hemolysis experiment - mouse, rat, or human? The cytotoxicity assay only tested one cell; it is recommended to include 2~3 additional cells to better evaluate the extract's cytotoxicity.

6.Why is the title of Figure 4 placed above the image? The image quality is poor, lacking clarity, and there is no indication of magnification. Additionally, it lacks a column chart for data statistics.

7.At a concentration of 62.5 μg/mL, the cell survival rate is only 70%, which means 30% of the cells have already died. How can this still be considered safe and non-toxic?

Author Response

Major comments

  1. The study evaluated the antifungal effects of Anacardium occidentale leaf ethanol extract against Candida albicans and auris, and it found that the extract caused changes in the morphology and structure of Candida albicans, as well as the induction of intracellular ROS accumulation. nonetheless, despite the preliminary identification of the extract's main components as flavonoids, particularly methanolic extracts containing glycosylated quercetin derivatives and other phenolic compounds, the authors did not actually isolate and purify the specific active compounds for their experiments. All biological assays were conducted using a crude mixture. Given the abundance of prior research on the antifungal properties of cashew leaf extracts, the study does not present a substantial leap in terms of novelty.

Answer:  The techniques used in our study are standardized and provide quantification of the extract as well as a possible mechanism of action. Therefore, our research does offer a deeper understanding of the antifungal activity, and it includes results on other species such as C. auris. Additionally, we are currently working on the purification of the molecule that we believe is responsible for this activity, which will be published in future works.

  1. While a slight decrease in spot density was observed for three mutants in the spot assay, there was no noticeable difference in ROS levels between the six mutated strains and the wild type after exposure to the extract in liquid culture conditions. The authors made extensive speculations to suggest a connection between the extract's antifungal activity and endoplasmic reticulum (ER) stress in fungi, but these remained mere conjectures, as they had not experimentally validated this hypothesis. To conclusively establish that the extract induces ER stress and mitochondrial dysfunction, additional experiments would be needed to substantiate the claim.

Answer: We acknowledge that additional experiments are required to conclusively establish the connection between the extract's antifungal activity and ER stress. However, despite the need for further validation, our data on membrane potential using rhodamine 123 are conclusive in suggesting mitochondrial dysfunction. Furthermore, our findings rule out the extract's activity on the deleted proteins in the mutants. This indicates that the high levels of ROS in the reticulum are strongly linked to mitochondrial dysfunction, highlighting significant communication between these two organelles in maintaining cellular homeostasis.

  1. Line 380-383 The author inferred "This damage is caused by disruptions in ergosterol and sphingolipid levels" solely based on transmission electron microscopy results, which would require further experimentation for validation.

Answer: Attend the reviewer comment, we change the paragraph

From line 380-383: “This damage is caused by disruptions in ergosterol and sphingolipid levels…”

To line 383-385: “suggesting that the A. occidentale leaf extract would be causing disturbances in ergosterol and sphingolipid levels (Iyer et al., 2022; Prasad & Singh, 2013) responsible for the porosity, fluidity and ultrastructure of the plasma membrane and cell wall..”

  1. In the ROS (Reactive Oxygen Species) experiment, the author chose amphotericin B as a positive control. Amphotericin B is a commonly used antifungal drug in clinical practice, and its bactericidal activity is significantly stronger than fluconazole. The MIC against Candida albicans is generally 0.25μg/mL, However, the author used a concentration of 4 μg/mL in their experiment. Why was such a high concentration chosen? Furthermore, amphotericin B primarily targets fungal cell membranes by binding to ergosterol, thereby disrupting membrane integrity for its fungicidal effect. Is it appropriate to use it as a positive control for oxidative damage? Additionally, in the spot assay (Figure 8b), amphotericin B was employed at a concentration of 8 μg/mL, which is far beyond the growth threshold for Candida. How can SC5314 still be observed to grow in Figure 8?

We appreciate your insightful comments and understand the need for clarity regarding our choice of amphotericin B concentrations.

Amphotericin B is a powerful fungicide with three well-documented mechanisms of action: the irreversible binding to ergosterol to form pores or channels in the membrane, and the sponge model to remove ergosterol, both of which are classic mechanisms. Additionally, its role in exacerbated ROS production is well documented, as demonstrated in DOI: 10.1128/AAC.03570-14.

Regarding the concentrations selected for the 2-hour spot assay, they were based on the same article:

Mesa-Arango, Ana Cecilia et al. “The production of reactive oxygen species is a universal action mechanism of Amphotericin B against pathogenic yeasts and contributes to the fungicidal effect of this drug.” Antimicrobial Agents and Chemotherapy, vol. 58,11 (2014): 6627-38. doi:10.1128/AAC.03570-14.

We hope this clarifies the rationale behind our experimental design details.

  1. Were the six heterozygous haploid deletion strains used in the article synthesized in the laboratory or procured from a company? Please provide a clear statement. Furthermore, the text should include an explanation for the selection of these particular strains

Answer: Attend the reviewer comment part of this information was included

 Line 89-90and six heterozygous haploid deletion C. albicans strains purchased from the library developed by Xu, Deming et al. 2007(Xu et al., 2007) .

Reference: Xu, D., Jiang, B., Ketela, T., Lemieux, S., Veillette, K., Martel, N., Davison, J., Sillaots, S., Trosok, S., Bachewich, C., Bussey, H., Youngman, P., & Roemer, T. (2007). Genome-Wide Fitness Test and Mechanism-of-Action Studies of Inhibitory Compounds in Candida albicans. PLoS Pathogens, 3(6), e92. https://doi.org/10.1371/journal.ppat.0030092

  1. Line 152  RPMI medium should be clearly indicated as RPMI 1640 medium.Is there a 100 well plate? The author is requested to confirm this detail.

Answer: Attend the reviewer comment part of this information was included

As mentioned in the paper, the 100-well plate called Honeycomb is a specific plate from the FP-1100-C Bioscreen device from Thermo Labsystems. This is different from the 96-well microplate for broth microdilution assay.

  1. In the MIC determination experiment, the author employed a positive control value that was derived from a previously known value. Where does this value come from? Reference or results from previous laboratory measurements? However, in reality, due to the different sources of reagents and operators used in the antibacterial activity experiment, positive controls need to be set every time, which is also a quality control requirement. Therefore, it is inappropriate for the author to use the previous data in this article.

Answer: We appreciate your feedback and the importance of setting appropriate positive controls for each experiment. The values used serve as a reference to verify the susceptibility profile of the strains against fluconazole. In each broth microdilution experiment, we included sterility controls for the sterile medium, growth controls without treatments, and controls with fluconazole. Additionally, the reference strain was always included. This approach ensures the reliability and reproducibility of our results, adhering to quality control requirements.

  1. The SEM experiment observed bacterial morphology after 24 hours of extract treatment, while TEM observed structures after only 2 hours. Why is there such a significant time difference in the setup?

Answer: the main objective of SEM is to provide the morphology patterns after exposure to a compound. Changes in the cell surface are usually associated with longer times necessary to demonstrate the presence of pores, channels, or cell permeability, which are the last steps in order to cell death. For its part, TEM is more associated with a response after a shorter time, usually used to see the effect on the organelles. In our study, it was associated with oxidative damage that was demonstrated after 2 hours. Additionally, published study protocols support the selected times for both SEM and TEM.

References:

SEM: Staniszewska, M et al. “Candida albicans morphologies revealed by scanning electron microscopy analysis.” Brazilian journal of microbiology : [publication of the Brazilian Society for Microbiology] vol. 44,3 813-21. 10 Dec. 2013, doi:10.1590/S1517-83822013005000056

TEM (Li et al., 2015, 2020)

Iyer, K. R., Robbins, N., & Cowen, L. E. (2022). The role of Candida albicans stress response pathways in antifungal tolerance and resistance. IScience, 25(3), 103953. https://doi.org/10.1016/j.isci.2022.103953

Li, Y., Chang, W., Zhang, M., Li, X., Jiao, Y., & Lou, H. (2015). Diorcinol D Exerts Fungicidal Action against Candida albicans through Cytoplasm Membrane Destruction and ROS Accumulation. PLOS ONE, 10(6), e0128693-. https://doi.org/10.1371/journal.pone.0128693

Li, Y., Shan, M., Zhu, Y., Yao, H., Li, H., Gu, B., & Zhu, Z. (2020). Kalopanaxsaponin A induces reactive oxygen species mediated mitochondrial dysfunction and cell membrane destruction in Candida albicans. PLOS ONE, 15(11), e0243066. https://doi.org/10.1371/journal.pone.0243066

Prasad, R., & Singh, A. (2013). Lipids of Candida albicans and their role in multidrug resistance. Current Genetics, 59(4), 243–250. https://doi.org/10.1007/s00294-013-0402-1

Xu, D., Jiang, B., Ketela, T., Lemieux, S., Veillette, K., Martel, N., Davison, J., Sillaots, S., Trosok, S., Bachewich, C., Bussey, H., Youngman, P., & Roemer, T. (2007). Genome-Wide Fitness Test and Mechanism-of-Action Studies of Inhibitory Compounds in Candida albicans. PLoS Pathogens, 3(6), e92. https://doi.org/10.1371/journal.ppat.0030092

  1. Which type of red blood cells were used in the hemolysis experiment - mouse, rat, or human? The cytotoxicity assay only tested one cell; it is recommended to include 2~3 additional cells to better evaluate the extract's cytotoxicity.

 Answer: Human red blood cells were used according to Evans, Brian C et al.(2013)

Regarding the cytotoxicity assay, we will take this recommendation for articles in progress

Reference: Evans, Brian C et al. “Ex vivo red blood cell hemolysis assay for the evaluation of pH-responsive endosomolytic agents for cytosolic delivery of biomacromolecular drugs.” Journal of visualized experiments: JoVE ,73 e50166. 9 Mar. 2013, doi:10.3791/50166

  1. Why is the title of Figure 4 placed above the image? The image quality is poor, lacking clarity, and there is no indication of magnification. Additionally, it lacks a column chart for data statistics.

Answer: The title of figure 4 was placed below the image. Thank you for the comments, We will consider this for our publication with the isolated molecule.

  1. At a concentration of 62.5 μg/mL, the cell survival rate is only 70%, which means 30% of the cells have already died. How can this still be considered safe and non-toxic?

Answer:

The extract at MIC concentration of 62.5 μg/mL showed 30% death in fibroblasts. Although it is not totally non-toxic and safe, it can be considered low toxicity (CC50= 1955 μg/mL) and within the parameters of acceptability. The selectivity index (SI) is an important measure to identify substances with promising biological activity and low levels of toxicity. It can be defined as the ratio of the toxic concentration of a sample against its effective bioactive concentration. This ratio must be >10. In this context, the CC50/MIC ratio for A. occidentale was 24, which confirms its level of acceptability.

Reference:

Indrayanto, G., Putra, G. S., & Suhud, F. (2021). Chapter Six - Validation of in-vitro bioassay methods: Application in herbal drug research. In A. A. Al-Majed (Ed.), Profiles of Drug Substances, Excipients and Related Methodology (Vol. 46, pp. 273–307). Academic Press. DOI: 10.1016/bs.podrm.2020.07.005

Njeru, S. N., & Muema, J. M. (2021). In vitro cytotoxicity of Aspilia pluriseta Schweinf. extract fractions. BMC research notes, 14(1), 57. DOI:10.1186/s13104-021-05472-4

Reference: Chahardehi, Amir-Modarresi, Arsad, Hasni, Ismail, Noor-Zafirah, & Lim, Vuanghao. (2021). Low cytotoxicity, and antiproliferative activity on cancer cells, of the plant Senna alata (Fabaceae). Revista de Biología Tropical, 69(1), 317-330.

Reviewer 2 Report

Unmasking Antifungal Might: Anacardium occidentale Leaf Extract Unleashes Mitochondrial Disruption, Thwarting Candida albicans with Heightened Reactive Oxygen Species, by Quejada LF, evaluates the potential of using natural products in the fight against fungal infections, in the context of antifungal resistance. The idea is not new, since there are numerous articles published on this subject, where Anacardium occidentale is evaluated as a potential treatment or intensifier of other antifungals.

The study presented is interesting, I have a few recommendations for improved clarity.

- title - why is in the title only C albicans, and C auris not? Since the manuscript is about C auris also

- line 24 - please add antifungal before cytotoxicity, otherwise it might be interpreted that it is about the fungi

- lines 63-75 - please move this to the beginning of the introduction. It is more logical that you discuss the problem first (fungal infections) and after the solution - natural products

- line 80 - it is written that the study evaluated only C albicans, but in the abstract it says C auris also - a little confusing

Author Response

Reviewer 2

Major comments

Unmasking Antifungal Might: Anacardium occidentale Leaf Extract Unleashes Mitochondrial Disruption, Thwarting Candida albicans with Heightened Reactive Oxygen Species, by Quejada LF, evaluates the potential of using natural products in the fight against fungal infections, in the context of antifungal resistance. The idea is not new, since there are numerous articles published on this subject, where Anacardium occidentale is evaluated as a potential treatment or intensifier of other antifungals.

The study presented is interesting, I have a few recommendations for improved clarity.

Details comment

- title - why is in the title only C albicans, and C auris not? Since the manuscript is about C auris also

Answer: The ROS accumulation, spots with mutants and loss of membrane potential assays were not carried out in C. auris

- line 24 - please add antifungal before cytotoxicity, otherwise it might be interpreted that it is about the fungi

Answer: Attend the reviewer comment part of this information was changed

 From: Antifungal drug resistance and cytotoxicity highlight the need for effective antifungal therapeutics…

To: Resistance to conventional antifungal drugs and toxicity associated with high doses highlight the need for effective antifungal therapies.

- lines 63-75 - please move this to the beginning of the introduction. It is more logical that you discuss the problem first (fungal infections) and after the solution - natural products

Answer: Attend the reviewer comment part of this information was changed

From line 63-75 - Annually, invasive fungal diseases (IFDs) are responsible for the admission of more than one million patients to health…

To: line 42-56: Annually, invasive fungal diseases (IFDs) are responsible for the admission of more than one million patients to health…

- line 80 - it is written that the study evaluated only C albicans, but in the abstract it says C auris also - a little confusing

Answer: Although the extract was tested on Candida auris, our main objective was to describe the mechanism of action of the extract on Candida albicans, information that had not been published until now. Given, the result we obtained in Candida albicans SC5314 (reference strain), was tested in a clinical strain of C. albicans and two clinical strains of Candida auris. However, oxidative damage assays were not carried out on these strains, the title of this work.

Reviewer 3 Report

In this manuscript, authors present the antifungal activity, antifungal action mode, and potential mechanism of Anacardium occidentale (Cashew Leaf) leaves against Candida albicans. This manuscript contains noteworthy information. Nevertheless, some comments are suggested to be considered:

1.  In the 2.4 Determination of minimum inhibitory concentration, which strains were used as control group?

2.  It better to show the possible mechanism of action in the New Figure 11. Moreover, many figures should be merged, such as Figure 10 and Figure 11.

3.  How about the potential mechanism of Anacardium occidentale (Cashew Leaf) leaves against C. auris?

4. Further research are needed to determine the effect of Anacardium occidentale (Cashew Leaf) leaves in vivo?

In this manuscript, authors present the antifungal activity, antifungal action mode, and potential mechanism of Anacardium occidentale (Cashew Leaf) leaves against Candida albicans. This manuscript contains noteworthy information. Nevertheless, some comments are suggested to be considered:

1.  In the 2.4 Determination of minimum inhibitory concentration, which strains were used as control group?

2.  It better to show the possible mechanism of action in the New Figure 11. Moreover, many figures should be merged, such as Figure 10 and Figure 11.

3.  How about the potential mechanism of Anacardium occidentale (Cashew Leaf) leaves against C. auris?

4. Further research are needed to determine the effect of Anacardium occidentale (Cashew Leaf) leaves in vivo?

Author Response

Reviewer 3

Major comments and details comment

In this manuscript, authors present the antifungal activity, antifungal action mode, and potential mechanism of Anacardium occidentale (Cashew Leaf) leaves against Candida albicans. This manuscript contains noteworthy information. Nevertheless, some comments are suggested to be considered:

  1. In the 2.4 Determination of minimum inhibitory concentration, which strains were used as control group?

Answer: We use fluconazole as an antifungal control and the strains C. krusei ATCC 6258 and C. parapsilosis ATCC 22019 as described the Clinical and Laboratory Standards Institute (CLSI) guidelines (M27-A3). We have these strains in our collection of microorganisms.

Reference:

M27-A3: Reference Method for Broth Dilution antifungal susceptibility testing of yeast. 2008.

  1. It better to show the possible mechanism of action in the New Figure 11. Moreover, many figures should be merged, such as Figure 10 and Figure 11.

Answer: It is impossible to merge figures 10 and 11 since figure 10 shows the loss of mitochondrial membrane potential and figure 11 shows permeabilization of the plasma membrane. two processes that are not necessarily correlated.

  1. How about the potential mechanism ofAnacardium occidentale (Cashew Leaf) leaves against C. auris?

Answer: Although the extract was tested on Candida auris, our main objective was to describe the mechanism of action of the extract on Candida albicans, information that had not been published until now. Given, the result we obtained in Candida albicans SC5314 (reference strain), was tested in a clinical strain of C. albicans and two clinical strains of Candida auris. However, oxidative damage assays were not carried out on these strains, the title of this work.

  1. Further research are needed to determine the effect of Anacardium occidentale(Cashew Leaf) leaves in vivo?

Answer: Thanks for the comments, we are working in this subject. In vivo assays are in progress. It will be part of a new paper.

Reviewer 4 Report

This study evaluated the antifungal potential of the ethanolic extract of Anacardium occidentale (Cashew Leaf) leaves against Candida albicans and C. auris. The antifungal activity was tested by the broth microdilution method and growth kinetic test. SEM and TEM were performed to further explore its antifungal action mode. Additionally, heterozygous knockout strains associated with resistance to oxidative stress were included in the study. Overall, these findings are exciting and I recommend possibly publishing after minor revision.

These findings are exciting and I recommend possibly publishing after minor revision, the issues are listed as follows.

 1. As shown in Figure 1a (Ultra-performance liquid chromatography (UPLC) chromatogram obtained from A. occidentale leaf extract at 350 nm), the content of Agathisflavone is relatively high. Did the author try to separate and purify Agathisflavone, or directly purchase ketones to study their bactericidal activity and mechanism of action? Although different compounds were tentatively identified based on their HRMS, UV spectra, retention time, and comparison with the data in the literature, there are still some unknown substances in the crude extract, which is more convincing if the purified substance can be used for research.

2. In the part of “2.2 Plant material and extraction”, whether the extraction method is a new method to explore or reference the literature method, it is suggested to add the literature citations.

3. The authors are advised to modify some formatting problems. For example, “performed, Additionally” (line 28), “,” should be “.”. “N-acetyl-L-cysteine” (line 95), the letters “N” and “L” should be italicized.

4. It is suggested to add literatures on Anacardium occidentale related research, especially in the areas of extraction, separation, and bioactive applications.

Author Response

Reviewer 4

Major comments

This study evaluated the antifungal potential of the ethanolic extract of Anacardium occidentale (Cashew Leaf) leaves against Candida albicans and C. auris. The antifungal activity was tested by the broth microdilution method and growth kinetic test. SEM and TEM were performed to further explore its antifungal action mode. Additionally, heterozygous knockout strains associated with resistance to oxidative stress were included in the study. Overall, these findings are exciting and I recommend possibly publishing after minor revision.

Details comment

These findings are exciting, and I recommend possibly publishing after minor revision, the issues are listed as follows.

  1. As shown in Figure 1a (Ultra-performance liquid chromatography (UPLC) chromatogram obtained from A. occidentale leaf extract at 350 nm), the content of Agathisflavone is relatively high. Did the author try to separate and purify Agathisflavone, or directly purchase ketones to study their bactericidal activity and mechanism of action? Although different compounds were tentatively identified based on their HRMS, UV spectra, retention time, and comparison with the data in the literature, there are still some unknown substances in the crude extract, which is more convincing if the purified substance can be used for research.

Answer: As part of the research approach for this first part, our interest was to determine the antifungal potential of the crude extract of A. occidentale. However, due to the interesting results obtained in this part, purification processes of the major compound are currently being carried out to verify its biological potential, and it is being considered for publication in another article. Additionally, no related flavonoid was purchased to study its activity. For the moment, focus is intended on conducting experiments on the major flavonoid, Agathisflavone.

  1. In the part of “2.2 Plant material and extraction”, whether the extraction method is a new method to explore or reference the literature method, it is suggested to add the literature citations.

Answer: Reference citations were added where the extraction method is relevant. This method has been standardized by our research group and is detailed in articles such as:

Romero, M. Á. V., Chitiva, L. C., Bravo-Chaucanés, C. P., Hernández, A. X., Parra-Giraldo, C. M., & Costa, G. M. (2023). Black-eyed Susan vine (Thunbergia alata): chemical and antifungal potential evaluation of an invasive plant species in Colombia. Universitas Scientiarum, 28(2), 217-229.

Bravo-Chaucanés, C. P., Chitiva, L. C., Vargas-Casanova, Y., Diaz-Santoyo, V., Hernández, A. X., Costa, G. M., & Parra-Giraldo, C. M. (2023). Exploring the Potential Mechanism of Action of Piperine against Candida albicans and Targeting Its Virulence Factors. Biomolecules, 13(12), 1729.

  1. The authors are advised to modify some formatting problems. For example, “performed, additionally” (line 28), “,” should be “.”. “N-acetyl-L-cysteine” (line 95), the letters “N” and “L” should be italicized.

Answer: Attend the reviewer comment part of this information was changed

(Line 28) From: scanning and transmission electron microscopy were performed, Additionally, heterozygous knockout strains

To: scanning and transmission electron microscopy were performed. Additionally, heterozygous knockout strains

(Line 95)From: rhodamine 123, N-acetyl-L-cysteine (NAC),

To: rhodamine 123, N-acetyl-L-cysteine (NAC),

  1. It is suggested to add literatures on Anacardium occidentale related research, especially in the areas of extraction, separation, and bioactive applications.

Answer: Some bibliographic references related to the A. occidentale extract were added, specifically focusing on extraction, separation, isolation, and its bioactive applications, such as:

Costa, A. R., Almeida-Bezerra, J. W., da Silva, T. G., Pereira, P. S., de Oliveira Borba, E. F., Braga, A. L., ... & Barros, L. M. (2021). Phytochemical profile and anti-Candida and cytotoxic potential of Anacardium occidentale L.(cashew tree). Biocatalysis and Agricultural Biotechnology, 37, 102192.

Ajileye, O. O., Obuotor, E. M., Akinkunmi, E. O., & Aderogba, M. A. (2015). Isolation and characterization of antioxidant and antimicrobial compounds from Anacardium occidentale L. (Anacardiaceae) leaf extract. Journal of King Saud University-Science, 27(3), 244-252.

Chaves, O. A., Lima, C. R., Fintelman-Rodrigues, N., Sacramento, C. Q., de Freitas, C. S., Vazquez, L., ... & Souza, T. M. L. (2022). Agathisflavone, a natural biflavonoid that inhibits SARS-CoV-2 replication by targeting its proteases. International Journal of Biological Macromolecules, 222, 1015-1026.

Costa, A. R., de Lima Silva, J. R., de Oliveira, T. J. S., da Silva, T. G., Pereira, P. S., de Oliveira Borba, E. F., ... & Barros, L. M. (2020). Phytochemical profile of Anacardium occidentale L. (cashew tree) and the cytotoxic and toxicological evaluation of its bark and leaf extracts. South African Journal of Botany, 135, 355-364.

Round 2

Reviewer 3 Report

No comments.

No comments.